# Single cell analysis reveals human cytomegalovirus drives latently infected cells towards an anergic-like monocyte state

Miri Shnayder[1], Aharon Nachshon[1], Batsheva Rozman[1], Biana Bernshtein[1], Michael Lavi[1], Noam Fein[1], Emma Poole[2], Selmir Avdic[3], Emily Blyth[3,4], David Gottlieb[3,4], Allison Abendroth[5], Barry Slobedman[5], John Sinclair[2], Noam Stern-Ginossar[1]*, Michal Schwartz[1]*

[1]Department of Molecular Genetics, Weizmann Institute of Science, Rehovot, Israel; [2]Department of Medicine, Addenbrooke's Hospital, University of Cambridge, Cambridge, United Kingdom; [3]Sydney Cellular Therapies Laboratory, Westmead, Sydney, Australia; [4]Blood and Bone Marrow Transplant Unit, Westmead Hospital, Sydney, Australia; [5]Discipline of Infectious Diseases and Immunology, Faculty of Medicine and Health, Charles Perkins Centre, University of Sydney, Sydney, Australia

*For correspondence:
noam.stern-ginossar@weizmann.
ac.il (NS-G);
michalsc@weizmann.ac.il (MS)

Competing interests: The authors declare that no competing interests exist.

**Abstract** Human cytomegalovirus (HCMV) causes a lifelong infection through establishment of latency. Although reactivation from latency can cause life-threatening disease, our molecular understanding of HCMV latency is incomplete. Here we use single cell RNA-seq analysis to characterize latency in monocytes and hematopoietic stem and progenitor cells (HSPCs). In monocytes, we identify host cell surface markers that enable enrichment of latent cells harboring higher viral transcript levels, which can reactivate more efficiently, and are characterized by reduced intrinsic immune response that is important for viral gene expression. Significantly, in latent HSPCs, viral transcripts could be detected only in monocyte progenitors and were also associated with reduced immune-response. Overall, our work indicates that regardless of the developmental stage in which HCMV infects, HCMV drives hematopoietic cells towards a weaker immune-responsive monocyte state and that this anergic-like state is crucial for the virus ability to express its transcripts and to eventually reactivate.

## Introduction

Human cytomegalovirus (HCMV) is a prevalent pathogen of the beta-herpesvirus family, infecting the majority of the human population worldwide (*Staras et al., 2006*). Following primary infection, HCMV persists through the lifetime of the host by establishing latency. In the latent state, no viral progeny is produced but the virus maintains the capacity to reactivate. Reactivation in immunocompromised individuals, such as transplant recipients and HIV patients, leads to severe illness and mortality (*Crough and Khanna, 2009*). Despite the significant health burden that accompanies HCMV reactivation from latency, to date there is no treatment that targets the latent stage and the processes governing latency and reactivation are far from fully understood.

HCMV has a wide cell tropism within its human host (*Mocarski and Shenk, 2013*), with most cell types supporting lytic replication (*Sinzger et al., 2008a*). In contrast, latent infection has so far been characterized in cells of the early myeloid lineage, including CD34+ hematopoietic stem and progenitor cells (HSPCs) and CD14+ monocytes in vivo (*Mendelson et al., 1996*; *Taylor-Wiedeman et al.,*

**eLife digest** Most people around the world unknowingly carry the human cytomegalovirus, as this virus can become dormant after infection and hide in small numbers of blood stem cells (which give rise to blood and immune cells). Dormant viruses still make their host cells read their genetic information and create viral proteins – a process known as gene expression – but they do not use them to quickly multiply. However, it is possible for the cytomegalovirus to reawaken at a later stage and start replicating again, which can be fatal for people with weakened immune systems. It is therefore important to understand exactly how the virus can stay dormant, and how it reactivates.

Only certain infected cells allow dormant viruses to later reactivate; in others, it never starts to multiply again. Techniques that can monitor individual cells are therefore needed to understand how the host cells and the viruses interact during dormant infection and reactivation.

To investigate this, Shnayder et al. infected blood stem cells in the laboratory and used a method known as single-cell RNA analysis, which highlights all the genes (including viral genes) that are expressed in a cell. This showed that in certain cells, the virus dampens the cell defenses, leading to a higher rate of viral gene expression and, in turn, easier reactivation.

Further experiments showed that the blood stem cells that expressed the viral genes were marked to become a type of immune cells known as monocytes. In turn, these infected monocytes were shown to be less able to defend the body against infection, suggesting that latent human cytomegalovirus suppresses the body's innate immune response.

The reactivation of human cytomegalovirus is a dangerous issue for patients who have just received an organ or blood stem cells transplant. The study by Shnayder et al. indicates that treatments that boost innate immunity may help to prevent the virus from reawakening, but more work is needed to test this theory.

---

*1991*; *von Laer et al., 1995*). Since CD14+ monocytes are short-lived cells it has been proposed that the latent reservoir resides in hematopoietic stem cells (HSCs) (*Slobedman et al., 2010*) and that latent monocytes support viral spread and persistence within the infected host (*Stevenson et al., 2014*). Latent cells in HCMV seropositive individuals are scarce and were estimated by PCR-driven in situ hybridization, at 1:10,000 to 25,000 with a copy number of 2 to 13 genomes per infected cell (*Slobedman and Mocarski, 1999*). Using highly sensitive methodologies, such as digital PCR, viral genomes were detected in less than half of seropositive individuals and viral load was estimated at less than 10 genomes in 10,000 cells in most individuals (*Jackson et al., 2017*; *Parry et al., 2016*). Due to the scarcity of HCMV-infected cells in the natural context, in vitro HCMV infection of primary cells, mainly HSPCs and monocytes were developed as models. The caveat of these systems is their heterogeneity and the possibility that they may represent dynamic differentiation states. Additionally, although both models are being widely used, the differences between them are not well understood.

It is becoming increasingly evident that the repertoire of viral genes expressed during latent infections is broader than initially appreciated (*Cheng et al., 2017*; *Schwartz and Stern-Ginossar, 2019*; *Shnayder et al., 2018*). Despite low expression of viral transcripts, a number of studies have described infection driven changes in host cells during HCMV latency (*Chan et al., 2010*; *Kew et al., 2017*; *Reeves et al., 2012*; *Slobedman et al., 2002*; *Smith et al., 2007*; *Smith et al., 2004*). Monocyte infection was proposed to promote differentiation to macrophages with specific polarization towards genes that mark M1 phenotype with some atypical attributes (*Chan et al., 2008*; *Smith et al., 2004*). On the other hand it was shown that the viral homolog of human interleukin-10, encoded by UL111A, polarizes monocytes into an anti-inflammatory M2 subset (*Avdic et al., 2013*). The UL7 viral protein was found to bind Fms-like tyrosine kinase three receptor (Flt-3R), inducing differentiation of HSPCs to monocytes and of monocytes to macrophages (*Crawford et al., 2018*). Finally, analysis of HCMV- infected HSPCs revealed reprogramming of HSPCs into immune-suppressive monocytes (*Zhu et al., 2018*). Thus, although it is clear that latent HCMV infection affects the differentiation state of infected HSPCs and monocytes, the nature of these effects is still enigmatic and controversial.

The studies to date examining host responses to latent HCMV infection focused on differences between infected and uninfected cells or on the effect of specific viral transcripts. Since the experimental systems for HCMV latency rely on primary immune cell populations, which are heterogeneous, and since infection is variable within the culture, it is likely that analyses of bulk populations could miss important signatures. Single cell-RNA-seq (scRNA-seq) provides a unique opportunity to depict viral and host heterogeneity simultaneously and thus to uncover functional connections between the cellular environment and viral gene expression. Indeed, several recent works that applied single cell transcriptomics, revealed novel insights into the complexity of the host response and cellular permissiveness for a number of well-studied viruses (*Douam et al., 2017*; *Drayman et al., 2019*; *Galinato et al., 2018*; *Rato et al., 2017*; *Russell et al., 2018*; *Steuerman et al., 2018*; *Wyler et al., 2019*; *Zanini et al., 2018*).

Using single cell RNA data, we analyzed host determinants that are associated with HCMV latency. In CD14+ monocytes, we identified two cellular cell surface markers, MHCII and its chaperon CD74, whose expression is inversely-correlated with viral transcript levels. We demonstrate these markers allow separating between cells harboring higher and lower viral transcript levels, that these differences are induced by HCMV infection and that the cells exhibiting higher viral transcript levels support more efficient reactivation of HCMV from latency. Using these markers, we show that latently infected cells display an intrinsic weaker immune response state that is important for the viral ability to express its genes and reactivate. Furthermore, analysis of 7500 infected HSPCs revealed very heterogeneous populations, but viral transcripts were only detected in cells expressing monocyte lineage markers, such as CD14. Remarkably, also in these HSPC-derived monocytes, higher viral transcript levels were associated with lower expression of CD74 and reduced immune response gene signature. Taken together, our findings highlight cell surface proteins associated with viral transcript levels and establish that both HSPC and monocyte infection models lead to establishment of HCMV latency in a similar anergic-like state of monocytic cells.

## Results

### Expression of host genes correlates with viral transcript levels

We have previously performed single cell RNA sequencing (scRNA-seq) on HCMV- infected CD14+ monocytes as a model for studying HCMV latency (*Shnayder et al., 2018*). We used the Massively Parallel RNA Sequencing (MARS-seq) platform (*Jaitin et al., 2014*; *Paul et al., 2015*) to analyze experimentally infected primary CD14+ monocytes at different days post infection (dpi). This approach provided high coverage of cellular and viral transcriptomes, spanning expression of more than 11,000 genes in 3,655 cells. We previously used this dataset to examine the viral transcriptome during HCMV latency, showing it largely mirrors a late lytic viral program, albeit at much lower levels of expression (*Shnayder et al., 2018*).

An inherent advantage of scRNA-seq is the ability to track viral and host expression within the same cell, thus allowing analysis of viral-host interactions while keeping information on cell-to-cell and infection heterogeneity. Viral transcript levels in the majority of the infected monocyte population was low to undetectable and projection of the cells using t-distributed stochastic neighbor embedding (t-SNE) demonstrated that the cell distribution is determined mainly according to host gene expression variations (*Shnayder et al., 2018*). Nevertheless, the cells are organized according to viral transcript levels (*Shnayder et al., 2018*) and *Figure 1A*), suggesting that much of the differences in host gene expression are associated with variation in viral transcript levels. We therefore calculated Spearman correlation coefficient between the expression of each host gene and the total number of viral transcripts across all cells (*Figure 1B*). 319 cellular genes showed significant positive or negative correlations with viral gene expression (Z score >2, *Figure 1—source data 1*). Since a key challenge in transcriptome analysis is to connect between snapshots of gene expression profiles and a functional outcome, we first focused on identifying cell surface markers that exhibited strong association with viral gene expression in the scRNA-seq data as these may allow us to enrich for cells with higher viral transcript levels within the population. Among the strongest co-varying genes were genes encoding for MHC class II (variants HLA-DRB1/HLA-DPB1/HLA-DRA/HLA-DQB1/HLA-DPA1/HLA-DQA1/HLA-DMA/HLA-DRB5/HLA-DQA2 and HLA-DMB) and the transcript of CD74, an MHCII chaperon as well as a cell surface receptor on its own (*Bergmann, 2012*). Indeed, across the infected

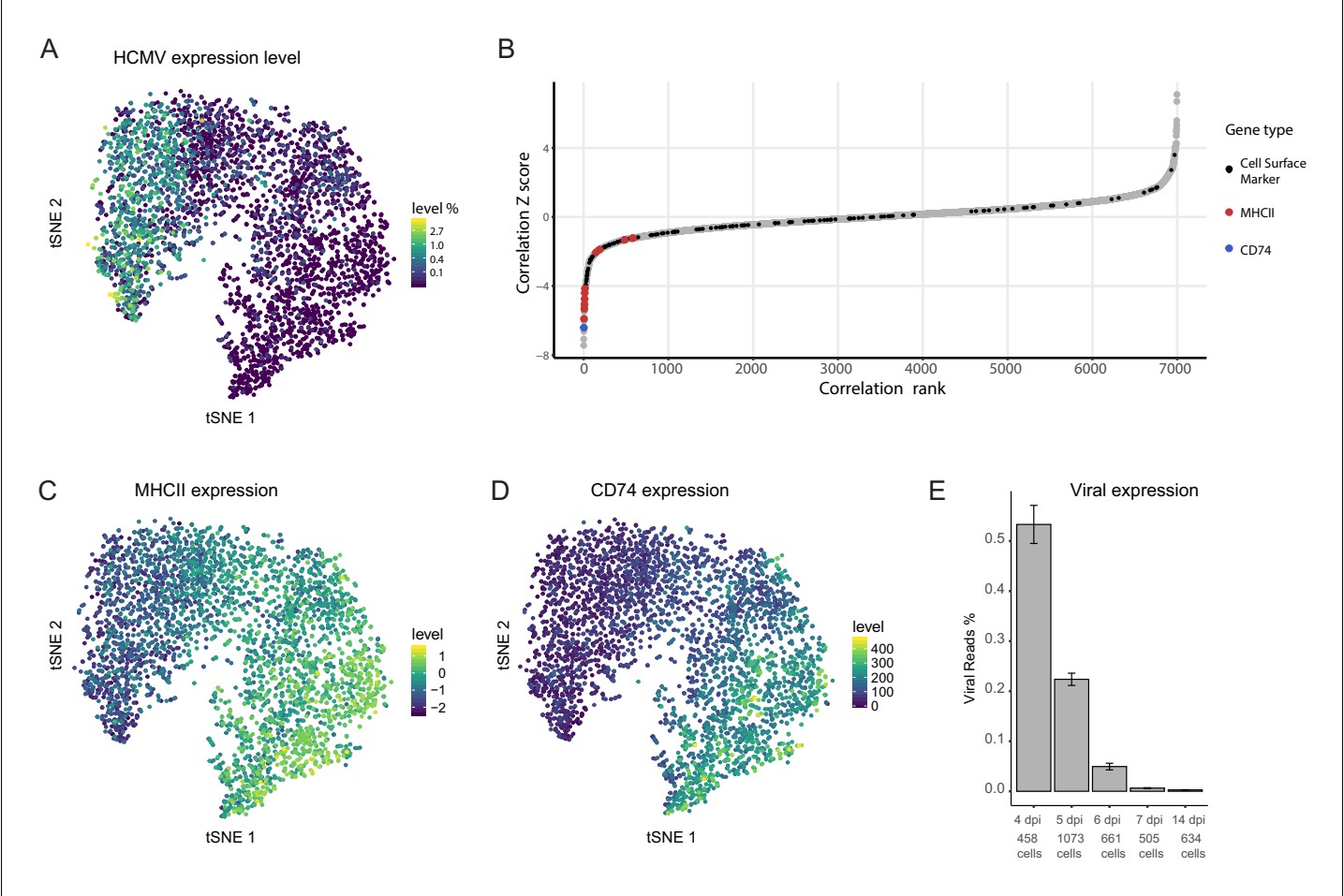

**Figure 1.** Cellular gene expression co-vary with viral transcript levels. (**A**) t-SNE plot of 3416 latently infected CD14+ monocytes based on host and viral gene expression (*Shnayder et al., 2018*), colored by percentage of HCMV reads per cell. (**B**) Distribution of Z scores of the Spearman correlation coefficients between host transcript levels and total HCMV transcript levels across a single population of infected CD14+ monocytes. Black dots mark cells surface markers coding transcripts, blue marks CD74 and red marks transcripts of MHCII (isoforms HLA-RB1/HLA-DPB1/HLA-DRA/HLA-DQB1/ HLA-DPA1/HLA-DQA1/HLA-DMA/HLA-DRB5/HLA-DQA2 and HLA-DMB, left to right), gray dots mark all other transcripts. (**C and D**) t-SNE plots of monocytes as presented in A colored coded by their expression levels of MHCII (HLA-RB1/HLA-DPB1/HLA-DRA/HLA-DQB1/HLA-DPA1/HLA-DQA1/ HLA-DMA/HLA-DRB5/HLA-DQA2 and HLA-DMB) (**C**) or CD74 (**D**) transcripts. (**E**) Percentage of viral reads measured in all single cells by days post infection (dpi). Error bars represent standard deviation across the single cells.

The online version of this article includes the following source data and figure supplement(s) for figure 1:

**Source data 1.** Correlation coefficients and Z scores between host transcript level and total HCMV transcript level across single infected CD14+ monocytes.

**Figure supplement 1.** High association between the correlation of host gene expression vs. viral gene expression and host gene expression vs. time post infection.

**Figure supplement 2.** CD74 and MHCII expression increases with culturing time of CD14+ monocytes.

monocyte population, the expression level of MHC class II (MHCII, *Figure 1C*) and CD74 (*Figure 1D*) showed clear inverse-correlation to viral transcript levels (*Figure 1A*).

## Cell-surface levels of CD74 and MHCII inversely-correlate with viral transcript levels

Viral gene expression levels in latent infection decreases with time in culture, likely due to continuous repression of the viral genome. This time related reduction in viral transcript levels is also apparent in our CD14+ monocytes scRNA-seq data (*Figure 1E*), thus the increase in CD74 and MHCII expression could be related to time in culture and to only indirectly inversely-correlate with viral transcript

levels (*Figure 1—figure supplement 1*). Indeed, we observed that the expression of both CD74 and MHCII increase over time in culture, both in infected and in uninfected cells (*Figure 1—figure supplement 2*). We therefore tested whether this inverse-correlation between CD74 and MHCII expression and viral transcript levels is upheld within a single infected population at single time points. To this end, we infected primary CD14+ monocytes with HCMV strain TB40/E-GFP (*O'Connor and Murphy, 2012*; *Sinzger et al., 2008a*). At 3 days post infection (dpi) the cells were FACS-sorted according to the cell-surface levels of CD74 and MHCII (*Figure 2A* and *Figure 2—figure supplement 1*). RT-qPCR analysis of viral gene expression in these populations confirmed that both CD74[low] and MHCII[low] monocyte populations express higher levels of viral transcripts compared to their high expressing counterparts (*Figure 2B* and *Figure 2—figure supplement 1*). Importantly, the association of higher viral transcript levels with lower cell-surface levels of CD74 was also maintained at 6dpi (*Figure 2—figure supplement 2*). We next examined if MHCII and CD74 are independent markers or are co-expressed and therefore mark the same population. mRNA expression analysis of CD74 and MHCII cells sorted according to the cell-surface levels of CD74 and MHCII as well as analysis of their co-expression in the scRNA-seq data, confirmed that the two markers are co-expressed and probably can be used to sort similar subpopulations within the infected cell population (*Figure 2C* and *Figure 2—figure supplement 3*). We therefore conducted all further experiments using CD74 based sorts. qPCR Analysis of viral DNA levels demonstrated that higher viral transcript

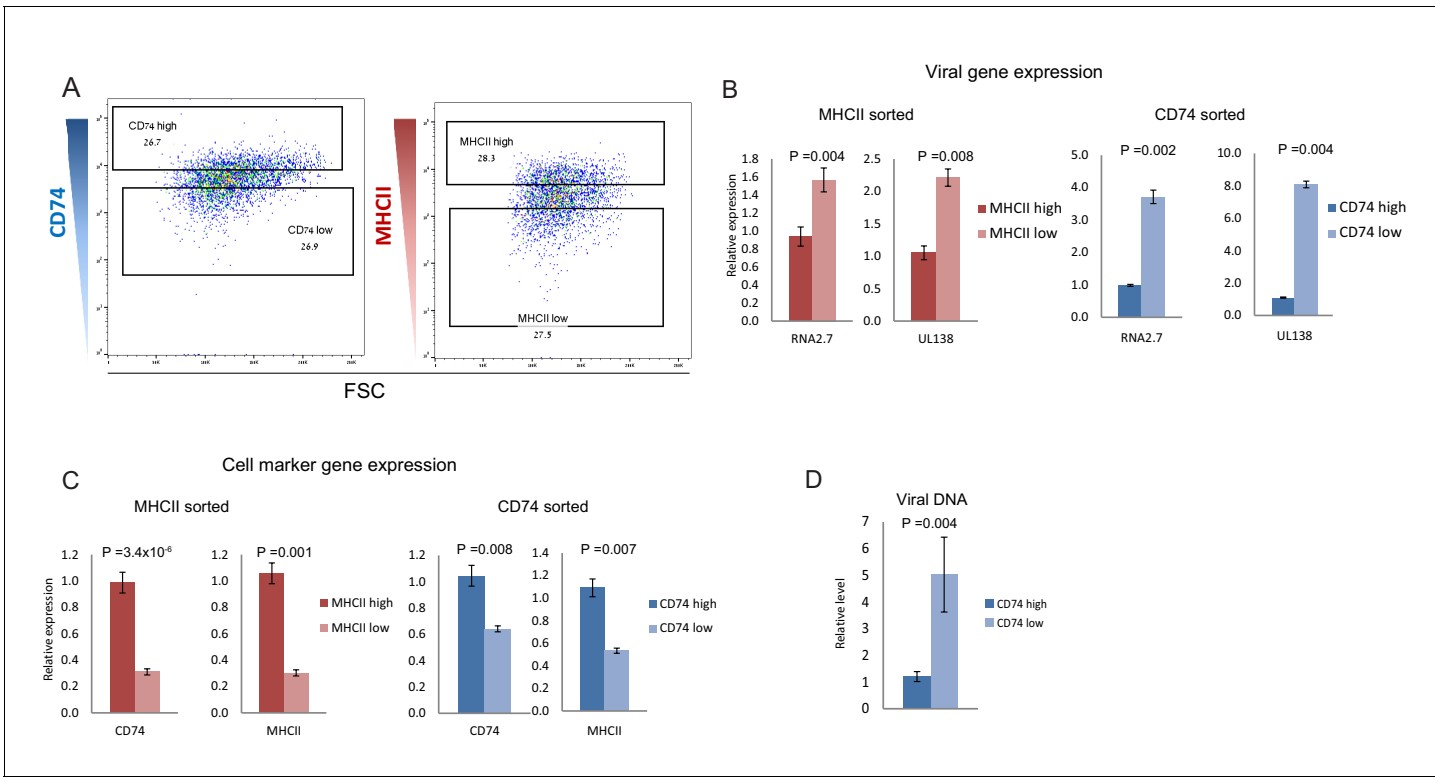

**Figure 2.** CD74 and MHCII cell-surface levels in HCMV- infected monocytes inversely-correlate with viral transcript levels. (**A**) HCMV- infected monocytes were FACS sorted according to cell-surface levels of CD74 or MHCII, at 3dpi. High and low gates were determined as highest and lowest 30% of the population, respectively. (**B and C**) Relative expression level of the viral transcripts RNA2.7 and UL138 (**B**) or MHCII and CD74 transcripts (**C**), as measured by RT-qPCR in HCMV- infected cells, sorted by either MHCII (left) or CD74 (right) cell-surface levels at 3dpi. (**D**) Relative abundance of viral DNA in HCMV- infected CD74[high] and CD74[low] monocytes, at 3dpi as measured by qPCR. Graphs show a representative experiment of 3 biological repeats, error bars reflect standard deviation of 3 measurements. P values as calculated by t-test are indicated.

The online version of this article includes the following figure supplement(s) for figure 2:

**Figure supplement 1.** CD74 and MHCII cell-surface levels in HCMV- infected monocytes inversely-correlate with viral transcript levels.
**Figure supplement 2.** Cell-surface CD74 levels in HCMV- infected monocytes at 6dpi inversely-correlate with viral gene expression.
**Figure supplement 3.** CD74 and MHCII genes are co-expressed in HCMV- infected monocytes.
**Figure supplement 4.** Surface expression distribution of CD74 does not change in uninfected and infected cell populations.

levels in CD74<sup>low</sup> monocytes are concurrent with higher abundance of viral genomes (*Figure 2D*) suggesting that differential loads of viral genome templates probably contribute to differential viral transcript levels and therefore to the effect on the host. Finally, we examined CD74 cell surface expression in infected and uninfected cells at different time points post infection but no major changes in CD74 distribution following infection were observed (*Figure 2—figure supplement 4*).

## Changes in CD74 and MHCII expression are induced by infection

There are two alternative explanations for the inverse-correlation between viral transcript levels and CD74 cell-surface levels, several days post infection with HCMV. The first possibility is that viral entry is more efficient in CD74<sup>low</sup> monocytes compared to CD74<sup>high</sup> monocytes, leading to more incoming viral genomes and higher viral transcript levels. In this case, differences in viral levels between CD74<sup>high</sup> and CD74<sup>low</sup> monocytes should be evident immediately following viral entry to the cells. An alternative option is that the differential expression of CD74 is driven by HCMV infection. In this case, the viral DNA and RNA levels in early stages of infection should be independent of CD74 cell-surface levels, and at later time points, higher load of virus leads to the observed differences in CD74 expression. To test these possibilities, uninfected freshly isolated CD14+ monocytes were FACS sorted based on CD74 cell-surface levels and then infected separately with TB40E-GFP. At 8 and 72 hr post infection (hpi) viral DNA and RNA were analyzed by qPCR. We confirmed that indeed the CD74<sup>high</sup> and CD74<sup>low</sup> sorted cells exhibited differences in CD74 transcript levels negating the possibility that the separation is only due to variations associated with the cell surface staining (*Figure 3A*). No significant differences between viral DNA load (*Figure 3B*) or viral transcript levels (*Figure 3C*) in CD74<sup>high</sup> and CD74<sup>low</sup> monocytes were observed at either 8 or 72hpi, indicating there are no major differences in the efficiency of viral entry between the two populations. Taken together, these results indicate that the observed variation in CD74 cell-surface levels is induced following HCMV infection.

## CD74<sup>low</sup> monocytes reactivate more efficiently

An important characteristic that defines latent infection is the ability of the virus to reactivate. Therefore, a key challenge in gene expression analysis is to connect between snapshots of gene expression profiles and the infection status of the cells, which is defined by this functional outcome. The identification of cellular cell surface markers that inversely-correlate with viral transcript levels provides a handle to connect between gene expression and the ability of the virus to reactivate. To test the association between viral transcript levels and reactivation efficiency, HCMV- infected primary CD14+ monocytes were sorted by CD74 cell-surface levels, and viral reactivation was induced by two complementary methods; cytokine driven differentiation to dendritic cells (DCs) (*Reeves et al., 2005*), followed by incubation with an indicator fibroblasts monolayer, or long term co-culturing with fibroblasts. Quantification of GFP positive plaques showed that reactivation was significantly more frequent in CD74<sup>low</sup> cells compared to CD74<sup>high</sup> cells, in both protocols (*Figure 4A and B*). Lysed samples of monocytes plated onto indicator fibroblasts produced no plaques, confirming that these infected cells did not produce any detectable infectious virus, consistent with latency prior to reactivation induction. These results demonstrate a functional difference in reactivation efficiency associated with cell-surface CD74 levels as well as with viral transcript levels and genome load.

## CD74 can be used to enrich for HCMV harboring CD14+ monocytes from viremic patients

Based on our experimental infection results, we next tested whether CD74 expression on CD14+ monocytes could be used as a cell surface marker that will allow enrichment of cells that contain HCMV genomes in healthy seropositive individuals. We used digital droplet PCR (ddPCR) to detect viral genome loads in CD14+ monocytes from seven seropositive donors. Despite using a highly sensitive platform (*Figure 5—figure supplement 1*) and detection of several positive events, testing 100,000–400,000 cells per donor did not provide us with levels of detection and reproducibility that allowed relative quantification (*Figure 5—figure supplement 2A and B* and *Supplementary file 1*). Recently, it was suggested that latent HCMV resides in a unique B7H4-positive monocyte subset (*Zhu et al., 2018*). To examine whether we can enrich for a cell population in which we can detect robust levels of HCMV genomes, we collected CD14+ monocytes from 5 healthy seropositive

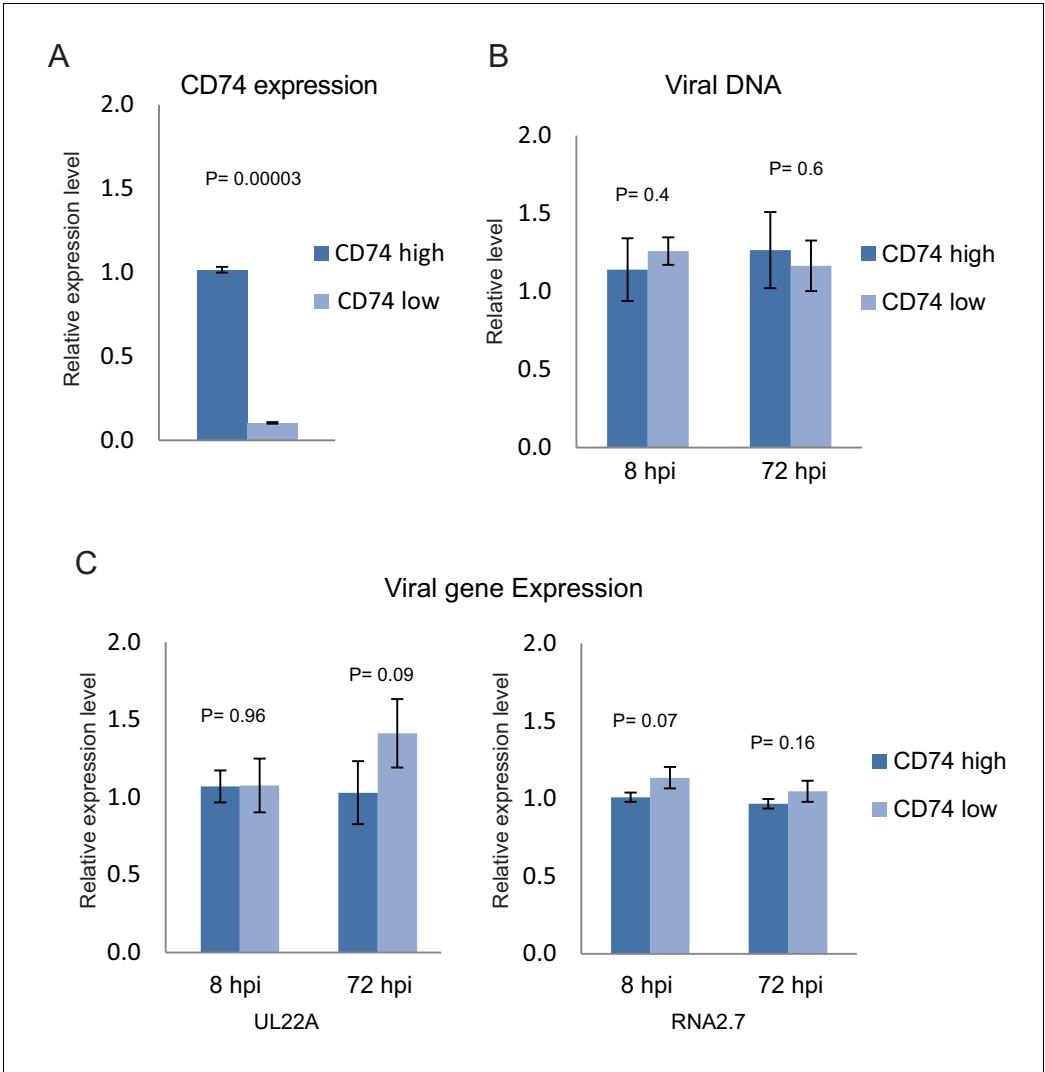

**Figure 3.** Changes in CD74 expression are induced by infection. Uninfected primary monocytes were FACS sorted according to cell-surface levels of CD74. Equivalent numbers of CD74high and CD74low cells were infected with HCMV and differences in CD74 RNA levels and in viral DNA and RNA levels between these two cell populations were assessed by qPCR. (A) Relative CD74 transcript levels in CD74high and CD74low cells at 8hpi. (B) Relative abundance of viral DNA in CD74high and CD74low cells at 8hpi and 72hpi. (C) Relative expression level of the viral transcripts UL22A and RNA2.7 in CD74high and CD74low cells as measured at 8hpi and 72hpi. Graphs show a representative experiment of 3 biological repeats, error bars reflect standard deviation of 3 measurements. P values were calculated by t-test.

individuals. In all five donors we could not detect a distinct population of B7H4-positive cells (*Figure 5—figure supplement 2C*). We nevertheless sorted the top 2% and the bottom 70% B7H4 stained cells and measured HCMV genome loads by ddPCR but did not detect higher levels of HCMV genomes in the top 2% B7H4 cells (*Figure 5—figure supplement 2D*). We therefore next analyzed CD14+ monocytes from seven hematopoietic stem cell transplant (HSCT) recipient samples in which HCMV viremia was detected, for the presence of viral genomes. In three of the tested samples no virus could be detected in CD14+ monocytes (*Figure 5—figure supplement 2E*). Monocytes from the four additional samples, in which viral genomes were detected (*Figure 5—figure supplement 2E*), were sorted according to CD74 cell-surface levels (*Figure 5A*). In agreement with our results in the experimental infection model, CD74 low monocytes were significantly enriched for the virus compared to CD74high monocytes, illustrating that also in this natural infection setting, CD74 expression can be used to enrich for monocytes carrying viral genomes (*Figure 5B and C*).

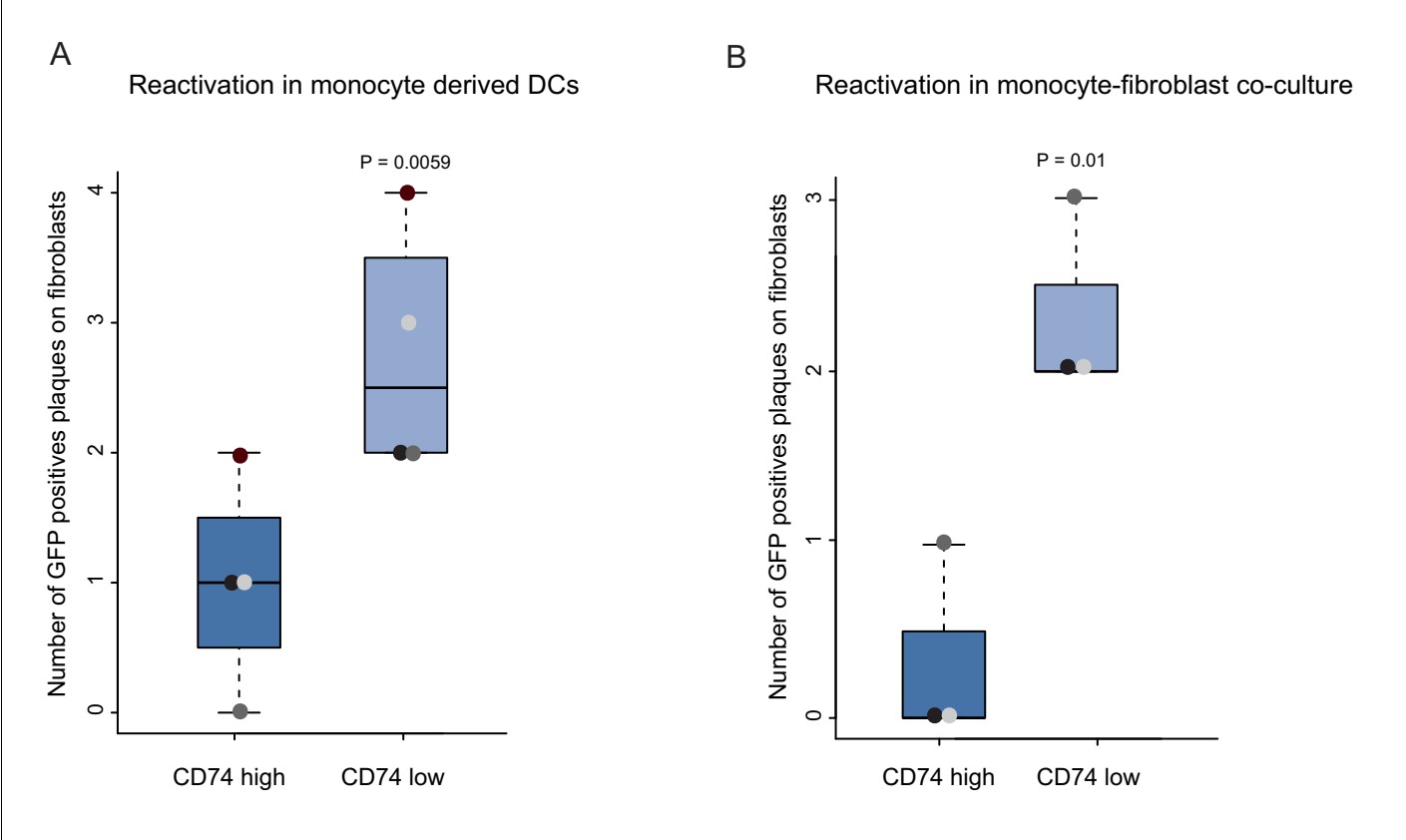

**Figure 4.** CD74[low] infected monocytes reactivate more efficiently. HCMV- infected CD14+ Monocytes were sorted according to cell-surface levels of CD74 at 3dpi. Equivalent numbers of cells were plated and (**A**) differentiated into DCs followed by co-culture with fibroblasts to quantify reactivation events or (**B**) directly co-cultured with fibroblasts to induce reactivation. Data ais shown as number of infectious centers formed by CD74[high] vs. CD74[low] cells. Means and error bars (reflecting standard deviation) were generated from 4 independent experiments (**A**) and 3 independent experiments (**B**). Dot colors indicate sets from the same experiment. P values as calculated by t-test are indicated.

## Cells with higher viral load express lower immune-responsive gene signatures

To further characterize the differences in host pathways associated with the variations in viral tran-script levels we conducted RNA-seq on latently infected monocytes, sorted according to their CD74 cell-surface levels. As was seen for single viral genes (*Figure 2B*), we found that on a genome-wide level, viral gene expression is higher in CD74[low] cells compared to CD74[high] cells (*Figure 6A* and *Figure 6—figure supplement 1A*). Furthermore, in accordance with our previous findings (*Shnayder et al., 2018*) the viral gene expression profile in these cells was correlated with late lytic profile (*Figure 6—figure supplement 1B*), but viral transcript levels were very low (~0.1% of mRNA reads originated from the virus). We performed differential gene expression analysis comparing between the CD74[low] and CD74[high] populations. This analysis revealed 113 differentially expressed cellular genes (FDR < 0.05, *Figure 6—source data 1*). Gene set enrichment analysis (GSEA) show that compared to CD74[low],CD74[high] monocytes, which exhibit lower viral transcript levels, are enriched for many immune response related pathways (*Figure 6B* and *Figure 6—source data 2*) including adaptive immune response and response to interferon gamma (*Figure 6C*). Since these cells were extracted from the same culture, these results suggest that monocytes carrying higher viral load are driven towards an intrinsic anergic-like phenotype. Reassuringly, many of the cellular genes and pathways exhibiting higher expression in CD74[high] cells were also inversely-correlated with viral transcript levels in the scRNA-seq data (*Figure 6D*, *Figure 6—figure supplement 2* and *Supplementary file 2*). The CD74[low] monocyte population, which exhibits higher viral transcript lev-els, was enriched in genes that are related to transcription and virus life cycle (*Figure 6B* and

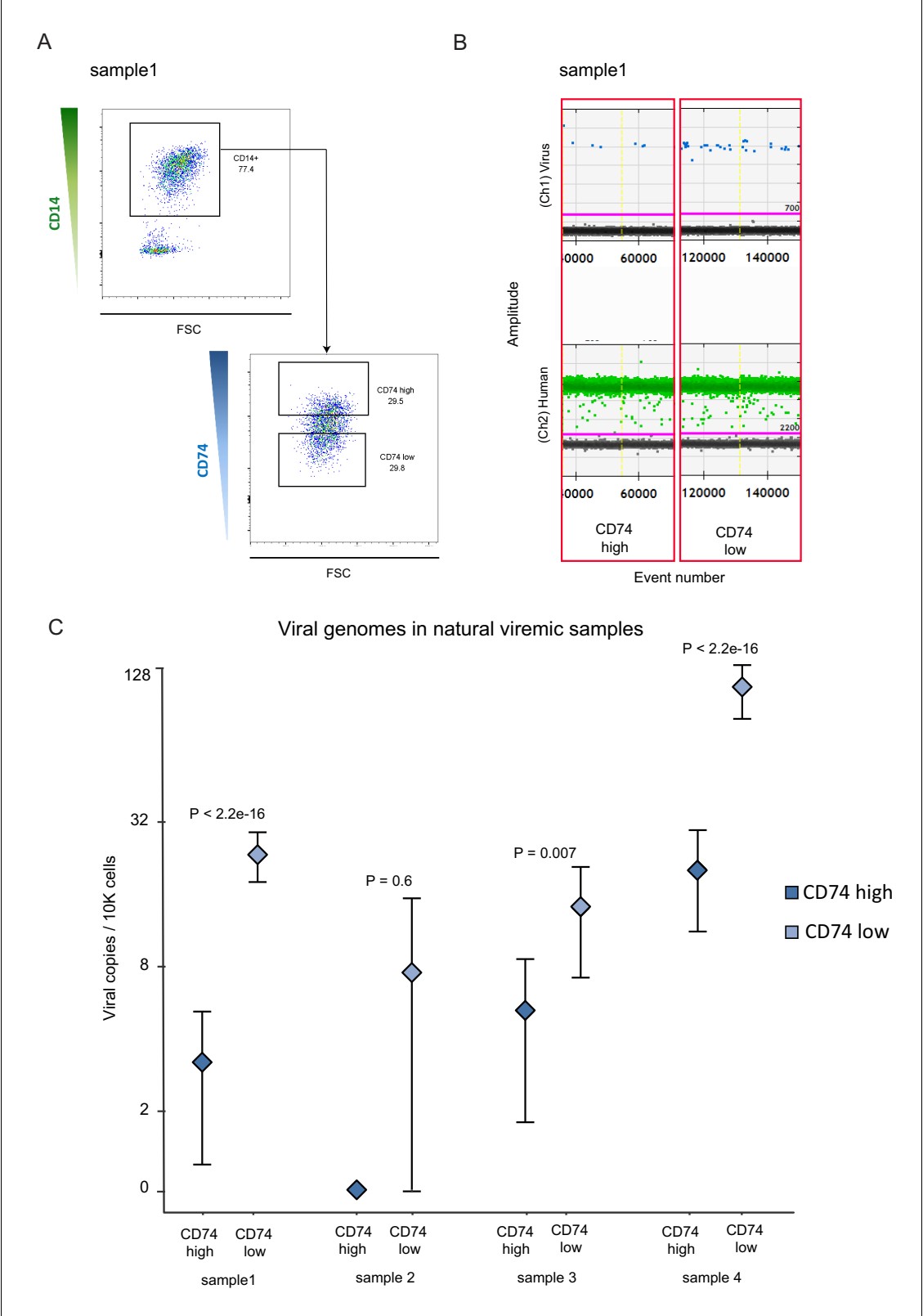

**Figure 5.** CD74 cell-surface levels allow enrichment of CD14+ monocytes carrying HCMV genomes from viremic patients. CD14+ monocytes from HSCT recipients with HCMV reactivation were sorted according to CD74 cell-surface levels. In total, 4 samples were tested, originating from 3 different donors (samples 1 and 2 were collected at different time points from the same donor), and viral genome abundance was measured by ddPCR. (A) A representative FACS sort of the cells. The highest and lowest 30% of the population were collected as CD74high and CD74low samples. (B) ddPCR

*Figure 5 continued on next page*

*Figure 5 continued*

results of two representative replicates from a single sample, separated by yellow vertical line. Upper panel shows detection of viral DNA, lower panel reflects detection of host genomes. The magenta line marks the threshold. (C) Quantification of viral genomes in CD74high and CD74low cells from four different samples, presented as copies per 10,000 cells. Graph reflects mean and 95% CV of poisson distribution, calculated from 5 technical replicates for each donor. P values as calculated by Fisher test are indicated.

The online version of this article includes the following figure supplement(s) for figure 5:

**Figure supplement 1.** Sensitivity of viral genomes detection by ddPCR.
**Figure supplement 2.** ddPCR analysis of natural samples.

*Figure 6—source data 2*). These pathways were not significantly correlated with viral expression levels in the single cell dataset, perhaps due to the sparser nature of our single cell measurements, which due to the sampling of limited number of cells may not detect some weaker effects.

To decipher the differentiation status of latent monocytes we compared our data to a recently published single cell analysis of lineage commitment during hematopoiesis (*Velten et al., 2017*). The genes that showed elevated expression in the CD74high cell population compared to the CD74low population were enriched for genes associated with monocyte lineage priming (*Figure 6E*), implying that the cells exhibiting higher viral transcript levels express less commitment markers and therefore may represent a less differentiated state. Significantly, this gene signature of monocyte lineage priming was also inversely-correlated with viral transcript levels in the scRNA-seq data (*Figure 6F*). To further investigate the cell differentiation state, we analyzed gene sets that were previously associated with M1 or M2 polarization (*Gerrick et al., 2018*) and found that genes that were more highly expressed in CD74high monocytes were significantly enriched for genes previously associated with M2 phenotype (Pval = $2.25 \times 10^{-8}$). This suggests that the infected CD14+ monocytes in our culture conditions may be polarized towards a cell-state associated with M2 phenotype, however higher viral transcript levels attenuates this differentiation trajectory.

## Inhibition of interferon signaling increases viral gene expression and reactivation

Since a main feature of CD74low monocytes is intrinsic reduced responsiveness to immune signals, we reasoned that this feature could contribute to the higher ability of the virus to express viral transcripts and eventually reactivate. To test this we examined whether inhibition of interferon signaling affects the expression level of viral transcripts and its ability to reactivate in infected CD14+ monocytes. We used ruxolitinib, a potent and selective Janus kinase (JAK) 1 and 2 inhibitor (*Lin et al., 2009*) that blocks the signaling downstream of interferon receptors. Treatment with ruxolitinib immediately after infection (3hpi) resulted in a considerable reduction in the levels of interferon-induced genes at 3 and 6dpi (*Figure 7A*) and in increased expression of essentially all viral transcripts (*Figure 7B*), in an overall uniform manner (*Figure 7—figure supplement 1*). In order to test how long after infection blocking interferon signaling still affects viral transcript levels, we applied ruxolitinib at different time points along infection and measured viral transcripts by RNA-seq. Interestingly, although the effect was smaller when the inhibitor was added later along infection, blocking interferon signaling still increased viral transcript levels even when added 3dpi (*Figure 7C*). These results therefore indicate there is continuous expression of viral genes and at 3dpi viral genes are still transcribed. However, the reduced effect of ruxolitinib with time also points that in addition there is gradual repression of viral gene expression as was also captured in our scRNA-seq analysis (*Shnayder et al., 2018* and *Figure 1E*). Importantly, blocking interferon signaling led to more efficient reactivation of the virus (*Figure 7D*). This demonstrates that interferon signaling has a major role in repression of viral gene expression in infected monocytes and that the anergic-like state of monocytes exhibiting higher viral transcript levels is likely important for latency maintenance and reactivation.

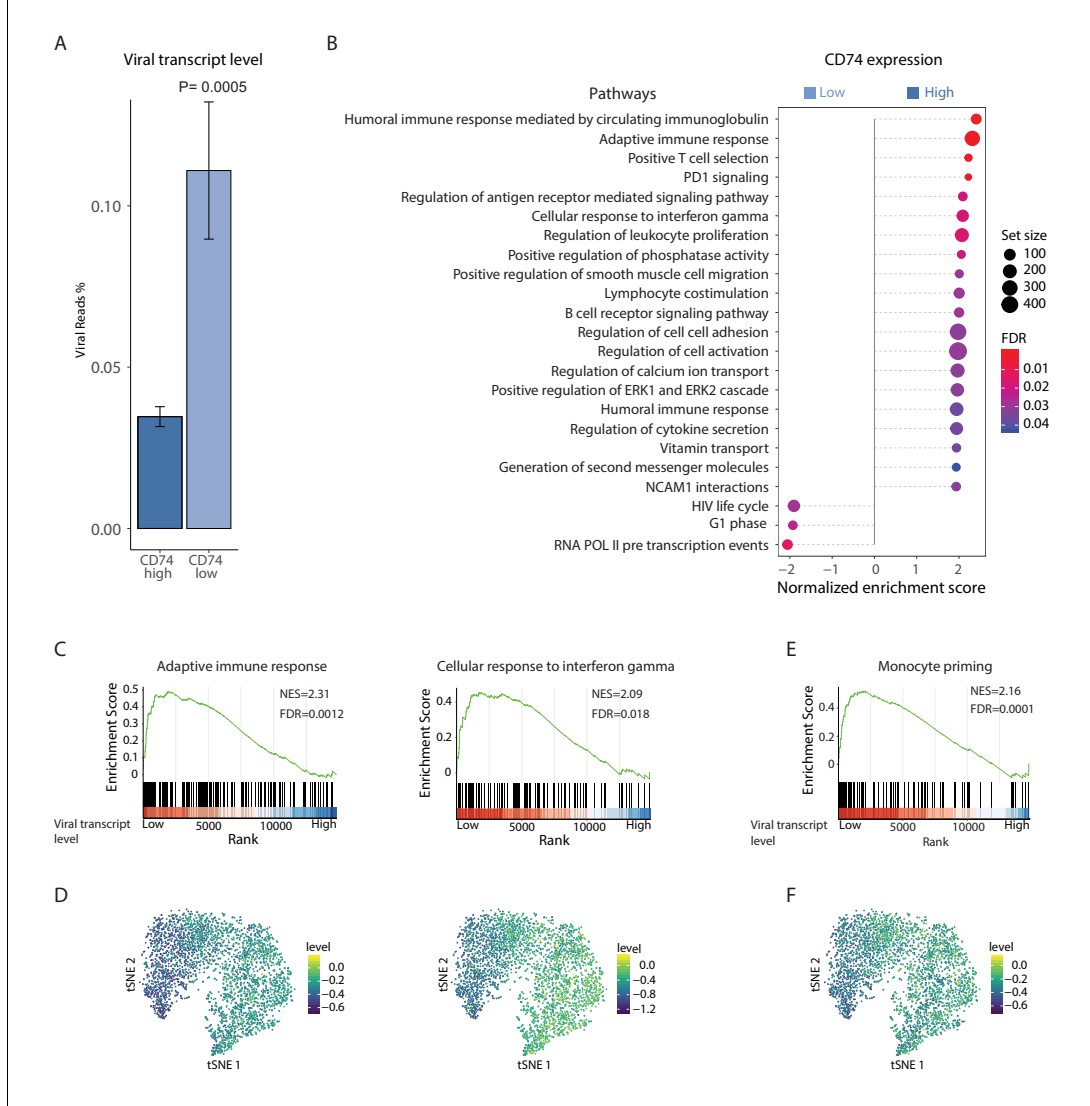

**Figure 6.** HCMV latency in monocytes is associated with reduced immune-response gene signatures. RNA-seq was performed on HCMV- infected CD14+ Monocytes that were sorted according to cell-surface levels of CD74 at 3dpi. (**A**) Normalized viral gene expression in CD74high and CD74low cells. P value, calculated using likelihood ratio test on logistic regression of viral reads, is indicated. (**B**) Summary of gene set enrichment analysis (GSEA) of differential expressed genes identified in RNA-seq analysis of CD74high and CD74low cells using annotated GO biological processes and Reactome pathways. (**C**) Representative pathways from GSEA of genes ranked by their differential expression between CD74high and CD74low cells. (**D**) tSNE plot of scRNA-seq of latent monocytes colored by expression level of the pathways shown in C. (**E**) Monocyte priming gene set from **Velten et al. (2017)** analyzed on GSEA. Genes are ranked by their differential expression between CD74high and CD74low monocytes. (**F**) tSNE plot of scRNA-seq of latent monocytes colored by the expression level of the monocyte priming gene set from **Velten et al. (2017)**.

The online version of this article includes the following source data and figure supplement(s) for figure 6:

**Source data 1.** DE analysis of RNA-seq on CD74high and CD74low cells.

**Source data 2.** GSEA of genes showing higher expression in CD74high compared to CD74low monocytes.

**Figure supplement 1.** Viral gene expression profile of infected CD14+ monocytes correlates between CD74high and CD74low cells and to late lytic profile.

**Figure supplement 2.** Comparison of changes detected in bulk RNA-seq and scRNA-seq data.

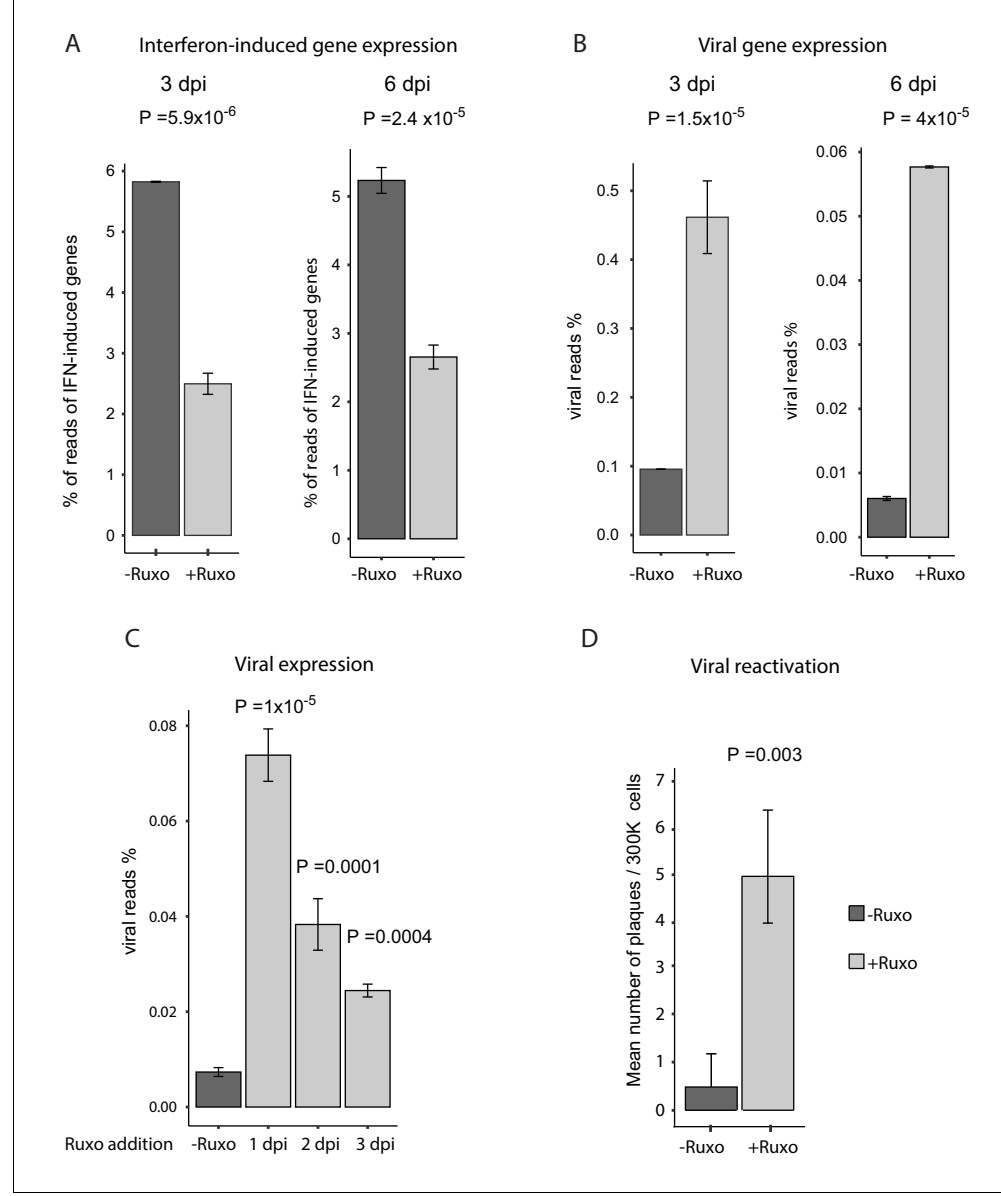

**Figure 7.** Inhibition of interferon signaling increases viral gene expression and promotes reactivation. (**A and B**) HCMV- infected monocytes were treated at 3hpi with interferon signaling inhibitor, ruxolitinib (ruxo), or left untreated, and analyzed for gene expression level, by RNA-seq. (**A**) Expression of interferon-induced genes at 3dpi (left) and 6dpi (right). (**B**) Expression of viral genes at 3dpi (left) and 6dpi (right). (**C**) Viral gene expression in HCMV- infected monocytes that were treated with ruxo at 1, 2, 3dpi or left untreated was measured by RNA-seq at 5dpi. P values for (**A-C**), calculated using likelihood ratio test on logistic regression of viral reads, are indicated. (**D**) HCMV- infected monocytes were treated at 3hpi with interferon signaling inhibitor, ruxolitinib (ruxo), or left untreated, and at 6dpi, equivalent numbers of monocytes were co-cultured with fibroblasts to induce reactivation. Viral reactivation in ruxo treated vs. untreated cells was assessed by count of GFP positive plaques formed on the fibroblasts. Means and error bars (reflecting standard deviation) were calculated from 2 independent biological repeats. P value as calculated using likelihood ratio test on Poisson regression of positive plaque events is indicated.

The online version of this article includes the following figure supplement(s) for figure 7:

**Figure supplement 1.** Uniform induction of expression across viral genes by inhibition of interferon signaling.

## Viral transcript levels in HCMV-infected CD34+ HSPCs are associated with priming towards the monocyte lineage and reduced immune-response

We have previously also performed scRNA-seq analysis of HCMV- infected CD34+ HSPCs at 4dpi (*Shnayder et al., 2018*). To gain insight into the effects of HCMV on HSPC differentiation, we inferred differentiation trajectories using Monocle, a strategy that allows placing single cells along a pseudotime continuum based on their gene expression (*Trapnell et al., 2014*; *Figure 8A*). The cells expressing HCMV transcripts were clustered in a region that exhibited the latest pseudotime (i.e the most differentiated state, *Figure 8B*). Remarkably, although we infected bone marrow derived CD34+ HSPCs, viral transcripts were expressed only in cells expressing monocyte lineage markers (*Velten et al., 2017*), such as IRF7, IRF8 (*Figure 8C*) and CD14 (*Figure 8D*). Taking into account that it was previously shown that HCMV infects multipotent hematopoietic stem cells (*Goodrum et al., 2004*), these results indicate that HCMV induces differentiation of infected HSPCs towards the monocyte lineage. Interestingly, more detailed analysis of the distinct group of cells that expressed monocyte markers, revealed that this group could be split into two close yet distinct clusters, with viral reads being detected in much more cells in one of the clusters. Interestingly, the cluster showing higher viral transcript levels exhibited lower expression of CD74 (Pval = $1.73 \times 10^{-95}$, *Figure 8E and F*) as well as gene signatures associated with lower immune response to interferon gamma and

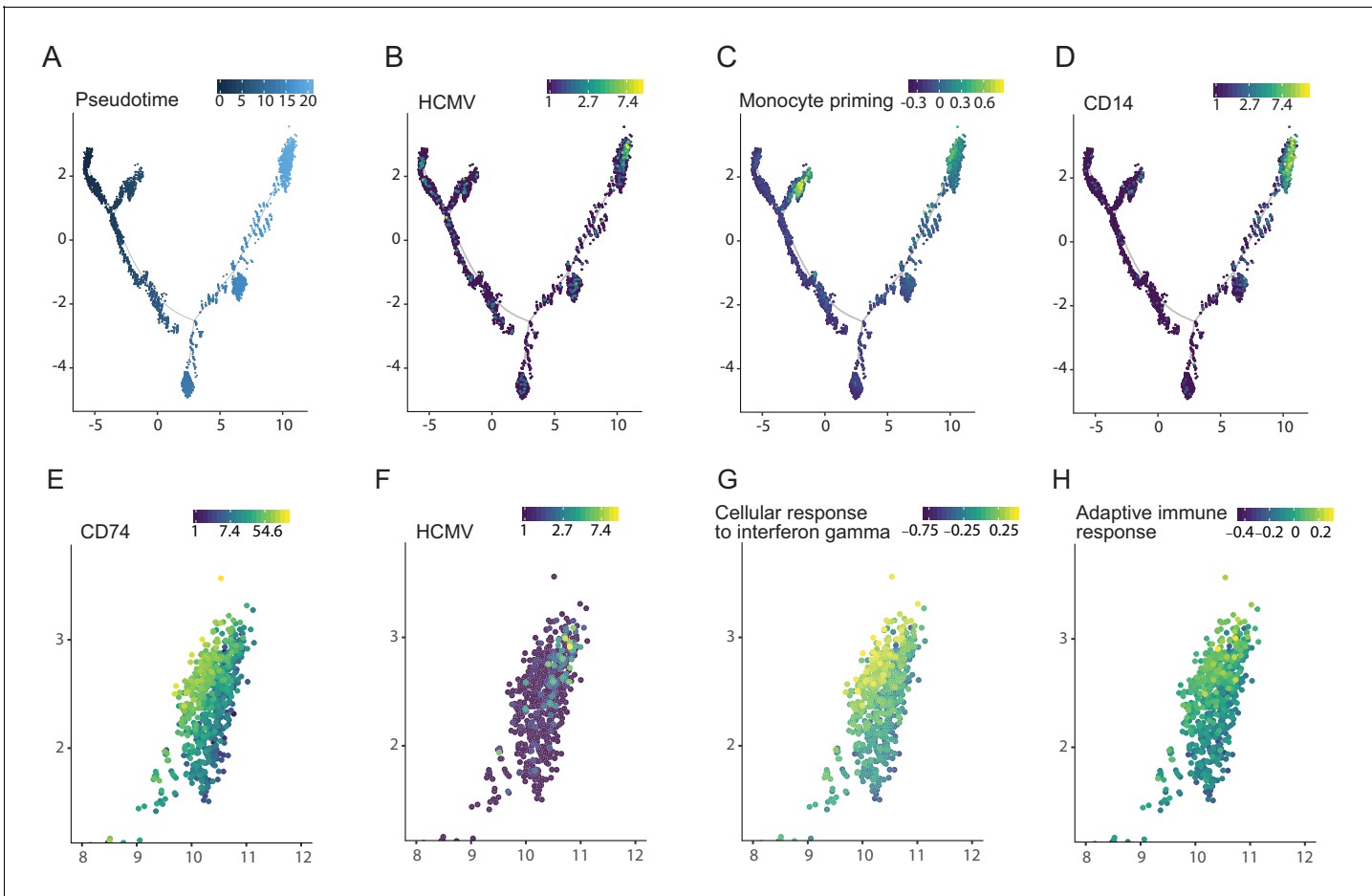

**Figure 8.** Viral transcript levels in HCMV-infected CD34+ HSPCs is associated with priming towards the monocyte lineage and with reduced immune-response. Pseudotime trajectory of single cell transcriptomes from HCMV- infected CD34+ HSPCs (n = 7,634, *Shnayder et al., 2018*) colored by pseudotime (A), viral transcript levels (B), expression of monocyte priming gene signature (*Velten et al., 2017*) (C) and CD14 expression (D). Zoom-in representation of the pseudotime trajectory in A-D, on the region exhibiting higher viral transcript level (top right group of cells), colored by expression of CD74 (E), viral transcript levels (F), expression level of the cellular response to interferon gamma pathways (G) and of the adaptive immune response pathway (H).

lower adaptive immune response (Pval <0.001 and Pval = 0.003, respectively, *Figure 8G and H*). Thus, cells within the HSPC population that exhibit the highest viral gene expression belong to the CD14+ monocyte cell lineage and are associated with similar anergic-like signatures as infected monocytes.

## Discussion

HCMV establishes latency in its host in progenitor cells of the myeloid system (*Mendelson et al., 1996*; *Taylor-Wiedeman et al., 1991*; *von Laer et al., 1995*). Nevertheless, in cell culture experimental systems it is apparent that not all cells have the ability to reactivate, indicating that there are variations in the levels or dynamics of latent infection. This heterogeneity means that bulk assays, comparing infected and uninfected cell populations can capture host responses to HCMV infection but likely miss specific responses in the group of cells in which latency is established, and will enable reactivation down the line.

Recent single cell RNA-seq data portray low-level expression of a broad spectrum of canonical viral lytic genes during HCMV latent infection of various cells of the hematopoietic system (*Galinato et al., 2018*; *Shnayder et al., 2018*). Our analysis of latent HCMV- infected CD14+ monocytes revealed a continuous population. Essentially, all of the cells were infected; however, they varied in the levels of viral transcripts. We exploited the host and viral heterogeneity, revealed simultaneously in scRNA-seq data, to look for associations between viral transcript levels and the human transcriptome. We found that among the genes that were highly inversely-correlated with viral transcript levels, is a group of genes encoding MHCII as well as the cell surface marker CD74, which serves as an MHCII chaperone. The finding that there are host genes that show specific correlation with viral transcript levels indicates that there is direct interaction between the host and the virus during latent infection- either there is a preference of the virus to infect specific cell types or that the virus actively affects the host transcriptome. Indeed, these markers could be used to enrich for cells with increased viral genome load and viral transcript levels. Importantly, this was supported by analysis of CD14+ monocytes from viremic patients, which also shows a clear inverse-correlation between CD74 cell-surface levels and viral genome load. These results are in line with previous findings showing that human, murine and rat cytomegalovirus down regulate the surface expression of MHCII molecules in cells of the myeloid lineage (*Baca Jones et al., 2009*; *Elder et al., 2019*; *Lee et al., 2011*; *Slobedman et al., 2002*; *Yunis et al., 2018*). In addition, CIITA, a transcription factor that regulates CD74 and MHCII, was shown to be down regulated by HCMV (*Lee et al., 2011*).

The inverse-association between the expression of CD74 and MHCII and viral transcript levels could be related to differences in permissivity for the virus. By cell sorting according to CD74 levels prior to infection, we show that this is likely not the case; instead, these changes seem to be induced by viral infection. Moreover, we see that higher viral transcript levels in CD74$^{low}$ cells is also accompanied by higher abundance of viral genomes, indicating that the level of viral transcripts is determined, at least initially, by the amount of incoming genomes, and this contributes to the extent of the effect on the host. Importantly, reactivation from HCMV latency happens in a very small population of cells even in experimental systems where the majority of cells are infected. We show here that CD74$^{low}$ monocytes, which carry higher viral transcript and viral genome levels, reactivate more efficiently. This indicates that the ability of monocytes to reactivate is associated with viral transcript expression and that the cells carrying higher viral loads in these models are functionally the latent cell population, as they are more likely to reactivate. Recently, B7H4 was suggested as a marker for monocytes with higher levels of HCMV genomes from HCMV seropositive individuals (*Zhu et al., 2018*), however, we could not detect expression of B7H4 mRNA in any of our RNA-seq samples. Moreover, by staining for B7H4, we could not identify a distinct B7H4 positive monocyte population and did not detect higher levels of HCMV genomes in the top 2% B7H4 sorted cells. According to available human datasets, B7H4 is indeed not expressed in healthy monocytes (*Blood Atlas, 2019*), however, these differences may stem from sampling of different donors or due to other variables related to the isolation of the cells.

Our analysis indicates that the cells that have higher expression of viral genes are less immune-responsive including reduced response to interferons. Since the CD74$^{high}$ and CD74$^{low}$ monocytes we analyzed grew in the same culture, they were exposed to the same immune-extrinsic signals. Hence, this difference is intrinsic to specific cells, and may be actively induced by the virus. This is

supported by a recent report showing that interferon induced genes are downregulated during latent infection of CD14+ monocytes (*Elder et al., 2019*). This anergic-like state is functionally related to the ability of the virus to express viral transcripts and to reactivate, as inhibiting the response to interferons resulted in increased expression of viral transcripts as well as increased reactivation efficiency. The importance of the role interferon plays in latent infection is further supported by the fact that CMV replication can be inhibited in otherwise permissive cells by treatment with interferon beta (*Dağ et al., 2014*). Similarly, extrinsic interferon stimuli inhibited HSV-1 reactivation from latency in neuronal cells (*Linderman et al., 2017*). Importantly, blocking interferon signaling increases viral gene expression even when done up to 3dpi, suggesting that there is continuous viral gene expression. However, the effect was smaller when the inhibitor was added later along infection, indicating a gradual repression of viral gene expression with time. Overall, these findings indicate that a major aspect of the maintenance of latency and of the ability to reactivate at the cellular level is a balance between opposing forces which also affect each other- the intrinsic immune response, specifically the interferon pathway, and viral transcript levels. Future work will have to delineate the mechanisms by which these immune response pathways control general viral transcription and how viral transcripts and viral proteins modulate the immune response during latent infection.

Previous works have described changes in the differentiation state of monocytes in response to infection or to specific viral genes (*Avdic et al., 2013*; *Chan et al., 2008*; *Smith et al., 2004*). The use of the CD74 marker, allowed us to focus on differentiation processes unique to the CD14+ monocytes with higher viral transcript levels and higher reactivation efficiencies. We show that higher viral transcript levels are associated with less expression of monocyte priming signature and lower expression of a gene signature associated with M2 phenotype. It was previously shown that the HCMV encoded IL-10 homolog polarizes monocytes towards an M2c macrophage phenotype (*Avdic et al., 2013*). It is possible that the total population is indeed polarized towards this direction; however, this process may be inhibited in cells with higher viral transcript levels. These findings may suggest that the virus promotes attenuation of differentiation processes that monocytes are undergoing in culture and in the blood (*Patel et al., 2017*).

Examining the differentiation state in HSPCs following infection is far more complicated than in monocytes, as CD34+ HSPCs are a mix of pluripotent cells as well as progenitor cells in different stages of lineage commitment (*Velten et al., 2017*). By applying single cell trajectory analysis, which allows recovery of gene expression kinetics of differentiating cells (*Trapnell et al., 2014*), and aligning our data with data of lineage commitment during hematopoiesis (*Velten et al., 2017*), we show that cells containing detectable viral transcripts belong largely to one specific population- cells primed towards the monocyte lineage. From our data, we cannot determine whether this is due to preferential infection by HCMV of monocyte lineage committed cells within the HSPC compartment or whether HCMV infects multipotent cells, which are then either preferentially skewed towards the monocytic lineage or viral gene expression is initiated only when cells start to differentiate in the monocytes lineage. However since it was previously demonstrated that multipotent cells are infected with HCMV (*Goodrum et al., 2004*), the latter options are more likely. Previous studies demonstrated that HCMV infection leads to an increase in monocyte markers (*Zhu et al., 2018*) and similar results were shown specifically for the viral gene UL7 (*Crawford et al., 2018*). Our results are in line with these studies and show directly and in an unbiased manner that several days post infection, viral gene expression can be found only in cells in the monocyte lineage. Moreover, within the monocyte lineage primed cells, the cells exhibiting higher viral transcript levels show the same markers and characteristics as we found when infecting CD14+ monocytes, primarily lower expression of CD74 and weaker immune responsiveness. This is especially interesting as it suggests that regardless of the developmental stage in which HCMV infects, the end point, a few days after infection, are cells with very similar characteristics.

Overall, we use here single cell data to pinpoint the characteristics of the latently infected cells in an unbiased manner. Our analyses indicate that HCMV drives human HSPCs and monocytes into a monocyte state characterized by anergic-like gene signature. These findings shed light on the characteristics of the latent reservoir, which may help in the effort of developing strategies to eradicate the latently infected cells.

# Materials and methods

**Key resources table**

| Reagent type (species) or resource | Designation | Source or reference | Identifiers | Additional information |
|---|---|---|---|---|
| Cell line (*Homo-sapiens*) | Primary human foreskin fibroblasts (HFF) | ATCC | Cat#: CRL-1634 | |
| Strain, strain background (HCMV virus) | TB40E-GFP | *O'Connor and Murphy, 2012*; *Sinzger et al., 2008a* | | |
| Antibody | anti-human APC-CD74 (Mouse monoclonal) | Miltenyi Biotec | Clone: 5–329 RRID:AB_2659190 | 1:200 |
| Antibody | anti-human PE-HLA-DR, DP, DQ (recombinant) | Miltenyi Biotec | clone: REA332 RRID:AB_2652177 | 1:200 |
| Antibody | anti-human FITC-CD14 (Mouse monoclonal) | Miltenyi Biotec | Clone: TÜK4 RRID:AB_244303 | 1:200 |
| Antibody | anti-human APC-B7H4 (Mouse monoclonal) | Biolegend | clone: MIH43 RRID:AB_2562580 | 1:200 |
| Antibody (isotype control) | APC-Mouse IgG1 isotype control (Mouse monoclonal) | Biolegend | clone: MOPC-21 RRID:AB_326443 | 1:200 |
| Commercial assay or kit | Human CMV HHV5 kit for qPCR using a glycoprotein B target | PrimerDesign | Cat#: Path-HHV5 | |
| Commercial assay or kit | HEX labeled RPP30 copy number assay for ddPCR | Bio-Rad | Cat#: dHsaCP1000485 | |
| Chemical compound, drug | Ruxolitinib | InvivoGen | Cat#: tlrl-rux | 4 µM |
| Peptide, recombinant protein | Human GM-CSF | Peprotech | Cat#: 300-03-20 | 1,000 U/ml |
| Peptide, recombinant protein | Human IL-4 | Peprotech | Cat#: 200-04-20 | 1,000 U/ml |
| Biological compound (*E. coli* 0111:B4) | LPS | Sigma | Cat#: L4391 | 500 ng/ml |
| Software, algorithm | R 3.5.1 | https://www.r-project.org/ | RRID:SCR_001905 | |
| Software, algorithm | Monocle 2.10.1 | http://cole-trapnell-lab.github.io/monocle-release/ | 2.10.1 | |
| Software, algorithm | DESeq2 1.22.2 | https://bioconductor.org/packages/release/bioc/html/DESeq2.html | RRID:SCR_015687 | |
| Software, algorithm | enrichplot | https://github.com/GuangchuangYu/enrichplot | | |
| Software, algorithm | GSEA 3.0 | http://software.broadinstitute.org/gsea/index.jsp | RRID:SCR_003199 | |
| Software, algorithm | MSigDB 6.2 | http://software.broadinstitute.org/gsea/index.jsp | RRID:SCR_016863 | Database used for GSEA |
| Software, algorithm | CellRanger 2.0.0 | https://support.10xgenomics.com/single-cell-gene-expression/software/pipelines/latest/what-is-cell-ranger | RRID:SCR_017344 | |
| Software, algorithm | lme4 1.1–21 | https://cran.r-project.org/web/packages/lme4/index.html | RRID:SCR_015654 | R package |

*Continued on next page*

*Continued*

| Reagent type (species) or resource | Designation | Source or reference | Identifiers | Additional information |
|---|---|---|---|---|
| Software, algorithm | Bowtie2 2.2.9 | http://bowtie-bio.sourceforge.net/bowtie2/index.shtml | | |
| Software, algorithm | Rtsne 0.15 | https://cran.r-project.org/web/packages/Rtsne/index.html | RRID:SCR_016342 | R package |

## Cells and virus stocks

Primary CD14+ monocytes were isolated from fresh venous blood, obtained from healthy donors, using Lymphoprep (Stemcell Technologies) density gradient followed by magnetic cell sorting with CD14+ magnetic beads (Miltenyi Biotec). The cells were cultured in X-Vivo15 media (Lonza) supplemented with 2.25 mM L-glutamine at $37^0$C in 5% CO2 (*Fortunato, 2014*). Primary human foreskin fibroblasts (HFF) (ATCC CRL-1634) were maintained in DMEM with 10% fetal bovine serum (FBS), 2 mM L-glutamine, and 100 units/ml penicillin and streptomycin (Beit-Haemek, Israel).

The TB40E virus containing an SV40-GFP tag (TB40E-GFP) was described previously (*O'Connor and Murphy, 2012*; *Sinzger et al., 2008b*). Virus was propagated by electroporation of infectious bacterial artificial chromosome (BAC) DNA into fibroblasts using the Amaxa P2 4D-Nucleofector kit (Lonza) according to the manufacturer's instructions. Viral stocks were concentrated by centrifugation at 26000xg, $4^0$C for 120 min. Infectious virus yields were assayed on THP-1 cells (ATCC TIB-202).

## Infection and reactivation procedures

For experimental latent infection, CD14+ monocytes were incubated with the virus for 3 hr, washed twice and supplemented with fresh media. To assess infection efficiency, a sample of the infected cell population was FACS analyzed for GFP expression at 3dpi. All experiments were conducted when there was a shift in GFP intensity of the entire population following infection, indicating all cells were infected. HCMV latency was validated by absence of GFP positive plaques on fibroblasts incubated with infected monocytes cell lysate.

For reactivation assays, infected monocytes were counted, plated and co-cultured with primary fibroblasts immediately or following differentiation into dendritic cells (DCs). DC differentiation was done by incubation of cells with granulocyte-macrophage CSF and interleukin-4 (Peprotech) at 1,000 U/ml for 5 days, followed by stimulation with 500 ng/ml of LPS (Sigma) for 48 hr (as previously described in *Cobbs et al., 2014*). Release of infectious virions was assayed by quantification of GFP positive plaques on the fibroblasts monolayer.

## Ruxolitinib treatment

Ruxolitinib (Ruxo) was added at a concentration of 4 uM, either immediately after infection (3hpi) or at later time points (1, 2, or 3dpi). Monocytes were washed to remove residual Ruxo before co-culturing with fibroblasts.

## Quantitative real-time PCR analysis

For analysis of RNA expression, total RNA was extracted using Tri-Reagent (Sigma) according to manufacturer's protocol. cDNA was prepared using qScript cDNA Synthesis Kit (Quanta Biosciences) according to manufacturer's protocol. For analysis of DNA levels, cells were lysed in a 1:1 mixture of PCR solutions A (100 mM KCl, 10 mM Tris–HCl pH 8.3, and 2.5 mM MgCl2) and B (10 mM Tris–HCl pH 8.3, 2.5 mM MgCl2, 1% Tween 20, 1% Non-idet P-40, and 0.4 mg/ml Proteinase K), for 60 min at 60°C followed by a 10 min 95°C incubation, as described in *Roback et al. (2001)*. Real time PCR was performed using the SYBR Green PCR master-mix (ABI) on the QuantStudio 12K Flex (ABI) with the following primers (forward, reverse):

UL 138 (GTGTCTTCCCAGTGCAGCTA, GCACGCTGTTTCTCTGGTTA)
UL22 (TTACTAGCCGTGACCTTGACG, CAGAAATCGAAGCGCAGCG)
RNA 2.7 (TCCTACCTACCACGAATCGC, GTTGGGAATCGTCGACTTTG)
CD74 (TGGAAGGTCTTTGAGAGCTGGATG, TTCCTGGCACTTGGTCAGTA)
MHCII-HLA-DQA1 (CTTCATCATCCAAGGCCTGC, CGGGCCAGAGAATAGTGCTA)

ANXA5 (AGTCTGGTCCTGCTTCACCT, CAAGCCTTTCATAGCCTTCC)

Viral DNA was quantified with RNA2.7 primers, host DNA was measured with the following primers (forward, reverse):

B2M (TGCTGTCTCCATGTTTGATGTATCT, TCTCTGCTCCCCACCTCTAAGT)

## Cell staining for flow cytometry and sorting

Cells were counted, and stained in cold MACS buffer (PBS, 5% BSA, 2 mM EDTA). Cell staining was done using the following antibodies: anti-human APC-CD74 (Clone: 5–329, Miltenyi Biotec), anti-human PE-HLA-DR, DP, DQ (clone: REA332, Miltenyi Biotec), anti-human FITC-CD14 (Clone: TÜK4, Miltenyi Biotec), anti-human APC-B7H4 (clone: MIH43, Biolegend), APC-Mouse IgG1 isotype control (clone: MOPC-21, Biolegend) according to manufacturer's instructions. Cells were analyzed and sorted on a BD FACSAriaIII.

## Detection of viral genomes by digital PCR

Detection of viral DNA in monocytes from natural latent samples was done using the QX200 droplet digital PCR system (Bio-Rad), using FAM labeled HCMV primer and probe: Human CMV HHV5 kit for qPCR using a glycoprotein B target (PrimerDesign); and HEX labeled RPP30 copy number assay for ddPCR (Bio-Rad), as previously described (*Jackson et al., 2017*). Calibration curve was ran in duplicate, using CMV positive control template (PrimerDesign). The limit of detection was 3 events per sample, with accuracy improved at 10 copies and higher (*Figure 5—figure supplement 1*). For sample preparation cells were counted, dry pelleted, and stored at −80°C prior to DNA extraction. DNA was extracted from the cell pellet in a 1:1 mixture of PCR solutions A (100 mM KCl, 10 mM Tris–HCl pH 8.3, and 2.5 mM MgCl2) and B (10 mM Tris–HCl pH 8.3, 2.5 mM MgCl2, 0.25% Tween 20, 0.25% Non-idet P-40, and 0.4 mg/ml Proteinase K), for 60 min at 60°C followed by a 10 min 95°C incubation, according to the description in *Roback et al. (2001)*.

## RNA library construction

RNA libraries were generated from samples of ~10,000 cells according to the MARS-seq protocol (*Jaitin et al., 2014*; *Keren-Shaul et al., 2019*).

## Sequencing and data analysis

RNA-Seq libraries (pooled at equimolar concentration) were performed in duplicates and sequenced using NextSeq 500 (Illumina), with read parameters: Read1: 72 cycles and Read2: 15 cycles.

Analysis of bulk MARS-seq of CD14+ monocytes, sorted according to the CD74 cell-surface levels, was done as described previously (*Shnayder et al., 2018*). The number of Unique Molecular Identifiers (UMIs) were: 976,294, and 947,474 for the CD74$^{high}$ samples, 902,150, and 844,872 for the CD74$^{low}$ samples (*Figure 6A–E*); 2,422,356 and 2,395,329 for the −Ruxo 3dpi samples, 2,187,290 and 925,362 for the +Ruxo 3dpi samples, 4,228,168 and 3,854,339 for the −Ruxo 6dpi samples and 3,449,188 and 1,900,301 for the +Ruxo 6dpi samples (*Figure 7A and B*); 2,076,741 and 1,858,871 for the +Ruxo at 1dpi samples, 1,914,289 and 1,163,469 for the +Ruxo at 2dpi samples, 1,554,298 and 1,685,834 for the +Ruxo at 3dpi samples, 2,004,859 and 2,450,039 for the −Ruxo control (*Figure 7C*).

Reads for gene expression and correlation analyses were normalized using DEseq2.

Based on the t-SNE plot of the CD14+ monocyte cells in *Shnayder et al. (2018)*, the latent cells were defined as all cells besides the small distinct group of cells that show high levels of viral reads (mean 9.5%). All the analyses in this paper include only the remaining 3,416 cells.

## Correlation and t-SNE coloring

For calculating the Spearman correlation between host genes and either viral expression levels or dpi, in each cell, the sum of viral reads was normalized to the total number of reads in the cell, and the number of reads of each host gene was normalized to the total number of host reads in the cell. The correlation was calculated across 1,448 cells for 6997 genes. Cells with no viral reads, or with less than 700 different host genes expressed, were omitted. Genes with total number of reads less

than 20 were ignored. The Z-score was calculated based on the mean and the standard deviation calculated over all 6997 genes.

Color coding of t-SNE (*Figure 1C*, *Figure 6D* and *Figure 7B*) or Monocle (e.g. *Figure 7C, G, H*) plots for expression levels of groups of genes was calculated according to the average relative expression as follows: First, genes with low level of expression, that is expressed in less than 50 cells, were omitted. Next, the number of host reads per cell was normalized so that all cells will include the same number of reads. Next, for each cell and each gene, the relative expression is calculated as $log_2\left(\frac{r_{ij}+1}{\bar{r_i}+1}\right)$ where $r_{ij}$ is the number of reads of gene $i$ in cell $j$ and $\bar{r_i}$ is the average number of reads of gene $i$ over all cells. Finally, for each cell the average $log_2$ relative expression over all genes in the group was calculated.

## Differential expression and enrichment analysis

The differential expression analysis was done with DESeq2 (version 1.22.2) (*Love et al., 2014*) using default parameters, with the number of reads in each of the samples as an input. The normalized number of reads according to DESeq2 were used for enrichment analysis using GSEA (version 3.0) (*Subramanian et al., 2005*). The genes sets that were used were GO biological process (c5.bp) and REACTOME (c2.cp.reactome) from MSigDB (version 6.2) (*Liberzon et al., 2011*) and the monocyte progenitor gene list from *Velten et al. (2017)*. The GSEA plots were created based on the GSEA output with the R package enrichplot. To calculate differential expression and pathway enrichment within the distinct group of cells that expressed monocyte priming markers, we used the graph-based clustering of the Cell Ranger software, stratifying this group of cells into two distinct clusters and GSEA (version 3.0) (*Subramanian et al., 2005*) for pathway enrichment. Differential expression and enrichment analysis were done on these two clusters. Enrichment of M1 and M2 associated genes was done by hypergeometric test taking all genes with mean expression ≥5 as background.

## Monocle analysis

CD34+ cells were ordered according to the predicted pseudo-time using Monocle (version 2.10.1) (*Trapnell et al., 2014*). The cells used in this analysis were filtered to have at least 1000 expressed genes and not more than 10,000 UMIs, while the genes were filtered to have mean expression greater than 0.1, and empirical dispersion greater than the global dispersion fit.

## Ethics statement

All fresh peripheral blood samples were obtained after approval of protocols by the Weizmann Institutional Review Board (IRB application 92–1). The study using HSCT recipient samples was approved by the Human Research Ethics Committee of the University of Sydney and the Western Sydney Local Health District. Informed consent was obtained from all study participants prior to enrolment in accordance with the Declaration of Helsinki.

# Acknowledgements

We thank the members of the Stern-Ginossar lab for critical reading of the manuscript. We thank Eain A Murphy for the TB40E-GFP virus strain. This research was supported by Infect-ERA (TANK-ACY), the European Research Council starting grant (StG-2014–638142) and a grant from the Abisch-Frenkel Foundation (18/WIS7) to NS-G, and by the Cambridge NIHR BRC Cell Phenotyping Hub and a grant from the British Medical Research Council (Grant G0701279) to JS. NS-G is incumbent of the skirball career development chair in new scientist. BS and EB were supported by a Biomed-Connect Grant from the University of Sydney.

# Additional information

### Funding

| Funder | Grant reference number | Author |
| --- | --- | --- |
| Seventh Framework Programme | Infect-ERA, TANKACY | Noam Stern-Ginossar |

| H2020 European Research Council | Starting grant (StG-2014-638142) | Noam Stern-Ginossar |
|---|---|---|
| Abisch-Frenkel Foundation | 18/WIS7 | Noam Stern-Ginossar |
| National Institute for Health Research | Cambridge NIHR BRC Cell Phenotyping Hub | John Sinclair |
| Medical Research Council | G0701279 | John Sinclair |

The funders had no role in study design, data collection and interpretation, or the decision to submit the work for publication.

### Author contributions

Miri Shnayder, Conceptualization, Formal analysis, Validation, Investigation, Visualization, Methodology, Writing - original draft, Writing - review and editing; Aharon Nachshon, Conceptualization, Software, Formal analysis, Validation, Investigation, Methodology, Writing - original draft, Writing - review and editing; Batsheva Rozman, Formal analysis, Investigation, Writing - review and editing; Biana Bernshtein, Formal analysis, Investigation, Methodology, Writing - review and editing; Michael Lavi, Resources, Investigation, Writing - review and editing; Noam Fein, Emma Poole, Investigation, Writing - review and editing; Selmir Avdic, Emily Blyth, David Gottlieb, Allison Abendroth, Resources, Writing - review and editing; Barry Slobedman, Conceptualization, Resources, Writing - review and editing; John Sinclair, Conceptualization, Investigation, Writing - review and editing; Noam Stern-Ginossar, Conceptualization, Formal analysis, Supervision, Funding acquisition, Investigation, Methodology, Writing - original draft, Writing - review and editing; Michal Schwartz, Conceptualization, Formal analysis, Supervision, Investigation, Methodology, Writing - original draft, Writing - review and editing

### Author ORCIDs

Emma Poole ![iD] http://orcid.org/0000-0003-3904-6121
Barry Slobedman ![iD] http://orcid.org/0000-0002-9431-6094
Noam Stern-Ginossar ![iD] https://orcid.org/0000-0003-3583-5932
Michal Schwartz ![iD] https://orcid.org/0000-0001-5442-0201

### Ethics

Human subjects: All fresh peripheral blood samples were obtained after approval of protocols by the Weizmann Institutional Review Board (IRB application 92-1). Informed written consent was obtained from all volunteers, and all experiments were carried out in accordance with the approved guidelines. The study using HSCT recipient samples was approved by the Human Research Ethics Committee of the University of Sydney and the Western Sydney Local Health District. Informed consent was obtained from all study participants prior to enrolment in accordance with the Declaration of Helsinki.

### Decision letter and Author response

Decision letter https://doi.org/10.7554/eLife.52168.sa1
Author response https://doi.org/10.7554/eLife.52168.sa2

## Additional files

### Supplementary files

• Supplementary file 1. ddPCR results of CD14+ monocytes from 7 donors Table showing the summary of ddPCR analysis of 8 samples from 7 healthy HCMV seropositive donors.

• Supplementary file 2. Correlation coefficients between CD74$^{high}$ enriched pathways (*Figure 6B*) and total HCMV transcript level across single infected CD14+ monocytes.

• Transparent reporting form

## Data availability

Sequencing data have been deposited in GEO under accession code GSE138838.

The following dataset was generated:

| Author(s) | Year | Dataset title | Dataset URL | Database and Identifier |
|---|---|---|---|---|
| Stern-Ginossar N, Shnayder M, Schwartz M, Nach-shon A, Rozman B, Bernshtein B, Lavi M, Fein N | 2019 | Single cell analysis reveals human cytomegalovirus drives latently infected cells towards an anergic-like monocyte state | https://www.ncbi.nlm.nih.gov/geo/query/acc.cgi?acc=GSE138838 | NCBI Gene Expression Omnibus, GSE138838 |

The following previously published dataset was used:

| Author(s) | Year | Dataset title | Dataset URL | Database and Identifier |
|---|---|---|---|---|
| Stern-Ginossar N, Shnayder M, Schwartz M, Nach-shon A, Boshkov A, Binyamin A, Maza I | 2018 | Defining the Transcriptional Landscape during Cytomegalovirus Latency with Single-Cell RNA Sequencing | https://www.ncbi.nlm.nih.gov/geo/query/acc.cgi?acc=GSE101341 | NCBI Gene Expression Omnibus, GSE101341 |

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
