## [Decision Letter]

**Acceptance summary:**

The human pathogen Human Cytomegalovirus (HCMV) causes a lifelong infection and achieves this through effective modulation of the host's immune response and establishment of a dormant state called latency. Reactivation from latency can cause life threatening disease. Our understanding of the molecular mechanisms of HCMV latency is far from complete. This study comprehensively characterizes, by applying single cell RNA-seq analysis, latency in monocytes and hematopoietic stem and progenitor cells (HSPCs). The authors identify monocyte cell surface markers that enable enrichment of latent cells which harbor higher viral transcript levels and can reactivate more efficiently. Importantly, these cells are characterized by reduced intrinsic immune responses, which is important for viral gene expression. For HSPCs latently infected with HCMV, viral transcripts could only be detected in monocyte progenitors, and these were also associated with reduced immune responses. In conclusion, this study shows that HCMV drives hematopoietic cells towards a weaker immune-responsive state, which is crucial for the virus to eventually reactivate and cause disease.

**Decision letter after peer review:**

Thank you for submitting your article "Single cell analysis reveals human cytomegalovirus drives latently infected cells towards an anergic-like monocyte state" for consideration by *eLife*. Your article has been reviewed by three peer reviewers, one of whom is a member of our Board of Reviewing Editors, and the evaluation has been overseen by Päivi Ojala as the Senior Editor. The following individual involved in review of your submission has agreed to reveal their identity: Lars Dölken (Reviewer #2).

The reviewers have discussed the reviews with one another and the Reviewing Editor has drafted this decision to help you prepare a revised submission. Overall, the reviewers were very impressed with the data and how it was presented.

Summary:

In their manuscript entitled "Single cell analysis reveals human cytomegalovirus drives latently infected cells towards an anergic-like monocyte state", the authors employ scRNA-seq to analyze determinants that are associated with HCMV latency. They identify surface expression of MHCII and the MHCII chaperon CD74 to be inversely correlated with viral transcript levels. They go on to show that these differences are not due to differences in virus entry but are rather induced upon virus entry. They show that cells exhibiting higher viral transcript levels support more efficient reactivation. The same phenotype was observed in HSPC-derived monocytes indicating that HCMV infection drives an immunologically anergic-like state of monocytic cells.

Essential revisions:

1) Figure 3: A useful experiment for Figure 2 or Figure 3 would be to quantify the increase in CD74 surface expression upon infection by FACS (histograms of CD74 signal at 0-14dpi). It would be interesting to see whether the distribution of CD74 surface expression is normally distributed before infection, and particularly whether it becomes bimodal/multimodal upon infection, as predicted from the scRNA-seq data. This would also support the conclusion from Figure 2, especially if the viral load could also be measured and would show lower values in cells with higher CD74 surface expression.

2) The authors show that viral gene expression in both CD14+ monocytes and HSPC essentially resembles late stage infection regarding the distribution of viral mRNA levels of the individual viral genes. As many viral genes act as immune evasins, it is not surprising that cells that express higher levels of these viral genes show signs of a counter-regulated IFNgamma response, namely a reduction in MHCII and CD74 surface levels. Furthermore, it is not surprising that this also augments virus reactivation. The million-dollar-question, however, is how it is possible that viral gene expression so closely matches the profile which is observed in late lytic infection despite no apparent DNA replication. The data obtained using ruxolitinib to inhibit IFN signaling provide strong evidence that the observed viral gene expression indeed results from de novo transcriptional activity and not from virion-associated RNA (the authors may want to state this in the Discussion). The key open question, however, is at what time after infection this viral gene expression occurs. It could either occur immediately after virus entry and then be more and more efficiently silenced, or it could be rather continuous thereby reflecting "true latent" viral gene expression. To clarify this important question, the authors should infect monocytes with HCMV and apply ruxolitinib at different time of infection and analyze the extent and expression profile of viral genes at 6dpi. This could e.g. comprise the following conditions: (i) ruxolitinib applied immediately with the virus, (ii) at 1dpi, (ii) at 2dpi, (iii) at 3 dpi, (iv) at 4 dpi and (v) at 5 dpi. Furthermore, they should apply ruxolitinib for the first three days and then replace the media with conditioned media from HCMV-infected monocytes. In case viral gene expression only occurs within the first 1-2 days post infection and is then efficiently silenced, treatment with ruxolitinib later on should not make any difference.

---

## [Author Response]

Essential revisions:1) Figure 3: A useful experiment for Figure 2 or Figure 3 would be to quantify the increase in CD74 surface expression upon infection by FACS (histograms of CD74 signal at 0-14dpi). It would be interesting to see whether the distribution of CD74 surface expression is normally distributed before infection, and particularly whether it becomes bimodal/multimodal upon infection, as predicted from the scRNA-seq data. This would also support the conclusion from Figure 2, especially if the viral load could also be measured and would show lower values in cells with higher CD74 surface expression.

We have examined CD74 surface expression in infected and uninfected cells along time (new Figure 2—figure supplement 4 and subsection “Cell-surface levels of CD74 and MHCII inversely-correlate with viral transcript levels”). The distribution of CD74 surface expression in infected cells does not significantly differ from what is seen in uninfected cells nor changed along infection. These results thus do not support two or several biologically distinct groups, rather a gradient of expression of CD74 which following infection is most likely affected by the innate immune response to infection (which increase CD74 expression) and by the effects of the virus (which counteracts this increase). Our results indeed show that the viral load (both genomes and transcripts) are lower in cells with higher CD74 surface expression (Figure 2B, D). However, in the MOI we have used (MOI=5) we do not think there are distinct groups of infected or bystander cells but rather a continuous (and probably random) distribution of viral load that inverse-correlates with a continuous distribution of CD74 expression.

2) The authors show that viral gene expression in both CD14+ monocytes and HSPC essentially resembles late stage infection regarding the distribution of viral mRNA levels of the individual viral genes. As many viral genes act as immune evasins, it is not surprising that cells that express higher levels of these viral genes show signs of a counter-regulated IFNgamma response, namely a reduction in MHCII and CD74 surface levels. Furthermore, it is not surprising that this also augments virus reactivation. The million-dollar-question, however, is how it is possible that viral gene expression so closely matches the profile which is observed in late lytic infection despite no apparent DNA replication. The data obtained using ruxolitinib to inhibit IFN signaling provide strong evidence that the observed viral gene expression indeed results from de novo transcriptional activity and not from virion-associated RNA (the authors may want to state this in the Discussion). The key open question, however, is at what time after infection this viral gene expression occurs. It could either occur immediately after virus entry and then be more and more efficiently silenced, or it could be rather continuous thereby reflecting "true latent" viral gene expression. To clarify this important question, the authors should infect monocytes with HCMV and apply ruxolitinib at different time of infection and analyze the extent and expression profile of viral genes at 6dpi. This could e.g. comprise the following conditions: (i) ruxolitinib applied immediately with the virus, (ii) at 1dpi, (ii) at 2dpi, (iii) at 3 dpi, (iv) at 4 dpi and (v) at 5 dpi. Furthermore, they should apply ruxolitinib for the first three days and then replace the media with conditioned media from HCMV-infected monocytes. In case viral gene expression only occurs within the first 1-2 days post infection and is then efficiently silenced, treatment with ruxolitinib later on should not make any difference.

As suggested by the reviewer, we conducted the proposed experiment. The results for applying ruxolitinib at different time points along infection showed that although the effect on viral gene expression are weaker the later the drug is added, there is still a significant increase in viral gene expression even at later time points of ruxolitinib addition (new Figure 7C, subsection “Inhibition of interferon signaling increases viral gene expression and reactivation” and Discussion, fourth paragraph).

This indeed supports, as the reviewer said, that there is continuous viral gene expression but the reduced effect of ruxolitinib also points that viral gene expression is gradually repressed. The results of applying the drug at the beginning of infection and then removing it 3dpi showed decreased effect on viral transcript levels compared to adding a new dose of ruxolitinib also at 72hpi, further supporting there is continuous expression of viral transcripts. However, since replacing the media also for untreated cells led to decreased viral transcripts levels, as shown in Author response image 1, these results are more complicated for interpretation and we did not include them in the manuscript.
